# Vulnerability of Privacy-Preserving Visual Localization against Diffusion-based Attacks

## Abstract

Driven by the increasing use of visual localization (VL) in AR/VR and autonomous systems, privacy-preserving localization is a critical societal necessity. Current VL systems rely on cloud-based 3D scene representation storage and client-side feature extraction, thus creating significant privacy risks. A privacy breach is framed as a malicious actor recovering privacy-preserving representations being sent from the client to the server. This paper therefore aims at finding out what can be recovered from these representations and comparing the multiple privacy-preserving solutions within the literature. We define privacy as the inability to recover personally identifiable information from image representations, acknowledging that general scene details do not inherently represent a privacy breach. We assess the degree of privacy of a representation by evaluating the amount of sensitive information it contains. To that end, we introduce a new privacy attack in which we train a diffusion model to reconstruct images through conditioning on different groups of privacy-preserving representations. We then measure what can be recovered in the images through a set of comprehensive experiments, which effectively act as a proxy to evaluate the degree of privacy of the initial representations. We apply this comprehensive evaluation protocol on different privacy-preserving representations and provide the first comparison between multiple branches of privacy-preserving visual localization methods. We plan on releasing code and trained checkpoints.

## 1 Introduction

Visual localization (VL), a core component of self-driving cars Heng et al. (2019) and autonomous robots Lim et al. (2015), refers to the task of estimating the 6DoF camera pose from which an image was captured. VL systems require a representation of the scene in which the pose can be estimated. Such representations may be a 3D Structure-from-Motion (SfM) point clouds (Sarlin et al., 2019; Humenberger et al., 2022), a database of images with known intrinsic and extrinsic parameters (Zhou et al., 2020; Sattler et al., 2019), the weights of neural networks (Kendall et al., 2015; Brachmann et al., 2017), neural radiance fields (Moreau et al., 2022; Chen et al., 2024) or 3D Gaussians representations (Pietrantoni et al., 2025a). A pose may be estimated either by extracting descriptors from an image, matching them with descriptors in the 3D scene representation, and solving a minimal problem or in a render and compare optimization scheme where query image representations are iteratively aligned with 3D scene representations.

With the growing adoption of localization services in AR/VR, autonomous vehicles, and mobile applications, safeguarding user data has become a critical priority. Therefore, privacy-preserving visual localization is a fundamental societal necessity to enable scalable, secure, and ethical deployment in real-world environments. In practice, visual localization systems rely on cloud services to store 3D scene representations, while the client extracts relevant image features from the target image to be localized before transmitting them to the server. Protecting both the 3D map and the 2D query data is equally vital, as vulnerabilities in either could compromise user privacy and system integrity. This paper primarily focuses on the latter, framing the privacy breach as an adversary recovering information transmitted between the client and server. Our attack is directly applicable to renderings of 3D models, and hence can be used to potentially recover private details from 3D representations. Yet, we do not explore this direction in this work but rather focus on 2D query data.

Following prior work Pittaluga et al. (2019); Chelani et al. (2021); Pietrantoni et al. (2023); Zhou et al. (2022); Chelani et al. (2025), we define privacy as the inability to retrieve personally identifiable information (*e.g.*, faces, license plates, pictures, documents, or texture details) from the image representation. We further assume that general scene details, including coarse geometry or semantic information, do not represent a privacy breach. This is because, in the context of visual localization, a minimum level of discriminative information is required to achieve accurate localization. Exposure of broad semantic or geometric information is therefore expected within a visual localization pipeline. Privacy, however, is inherently subjective and context-dependent, influenced by environmental factors, legal frameworks, and individual values. As such, the primary challenge lies in quantifying privacy and identifying effective mechanisms for it.

We build upon the seminal work from Pittaluga et al. (2019), which derives a privacy attack designed to reconstructing images from the high dimensional descriptors typically used for pose estimation. These descriptors contain a massive amount of privacy sensitive information that can easily be recovered through the privacy attack. In this context, privacy is assessed via a proxy by evaluating the amount of information recovered in the reconstructed image. In contrast to Pittaluga et al. (2019), which employs a feedforward convolutional network (CNN) to regress RGB values for the privacy attack, we propose a novel diffusion-based approach that addresses some of the limitations of the prior method. Concretely, the feedforward convolutional approach in (Pittaluga et al., 2019) is ill-suited for inverting sparse privacy-preserving representations and struggles to account for uncertainty, resulting in suboptimal inversions. Instead, our diffusion-based privacy attack optimizes a denoising network to iteratively reconstruct images, conditioning the denoising process on privacy-preserving representations. The conditioning is achieved at two levels: locally, by concatenating pixel-level representations, and globally, by injecting structural information via a graph neural network. This multi-scale conditioning enables a more robust and comprehensive framework, allowing more accurate inversion even from sparse representations.

We also extend the evaluation framework proposed in Pittaluga et al. (2019), where to evaluate the privacy only image-level metrics are applied on the reconstructed images. We argue that these metrics fail to adequately capture the extent of leaked privacy-sensitive information, thus only partially reflecting the true level of privacy risk. To address this limitation, we follow the approach in (Pietrantoni et al., 2025b), where privacy is characterized through the descriptive capacity of advanced vision-language models (VLMs). We believe these models are better suited for quantifying privacy, as they can capture fine-grained details with comprehensive class granularity. This capability enables a more nuanced understanding of privacy risks compared to predefined object detectors, which were proposed as privacy metrics in (Pietrantoni et al., 2023).

In addition to the above metrics, *i.e.*, perceptual visual similarities and VLM-based description similarities, we introduce a new privacy evaluation metric based on diffusion models. We leverage the probabilistic nature of diffusion models and evaluate the variability of the reconstruction from different random initial states. This proxy is a powerful way of measuring the amount of information contained in image representations.

In the last few years, multiple works have been proposed to tackle privacy-preserving visual localization (PPVL). These approaches can be classified into two main categories, *geometric obfuscation methods* (Speciale et al., 2019a;b; Shibuya et al., 2020; Geppert et al., 2020), which obfuscate the geometry of the scene while keeping image representations (descriptors) intact for downstream visual localization, and *descriptor obfuscation methods* (Dusmanu et al., 2021; Ng et al., 2022; Pittaluga & Zhuang, 2023; Pietrantoni et al., 2023; Wang et al., 2024; Pietrantoni et al., 2025a), which use image representations that contain little to no privacy sensitive information. However, these methods all use different privacy evaluation protocols and make independent claims of their level of privacy. In this work, we are the first to provide a shared benchmark and privacy evaluation protocol based on a comprehensive set of metrics for these methods. For reproducibility purposes and to foster evaluation of others representations, we plan on releasing code and trained checkpoints.

In summary: 1) We propose a new powerful diffusion-based inversion method, used as a privacy attack to reconstruct images from privacy-preserving representations. 2) We offer an in-depth comparisons of multiple privacy-preserving visual localization methods through a new unified and complete privacy evaluation protocol.

## 2 RELATED WORK

**Privacy-preserving visual localization and mapping** aim to mitigate the risk of inversion attacks, where attackers reconstruct recognizable images from point clouds, stored descriptors, or the representations exchanged between a client and a server. As demonstrated by Pittaluga et al. (2019), given a set of local descriptors with their 2D positions, it is possible to recover the original image via an inversion attack. Pittaluga et al. (2019) also show that their method can be extended to enable the reconstruction of detailed and recognizable scene content even from sparse 3D point clouds, thereby showing that storing such point clouds poses a significant privacy risk. To prevent such attacks and increase the degree of privacy, geometric obfuscation methods have been introduces to conceal the position of 2D/3D points. In particular, Speciale et al. (2019a;b) and Shibuya et al. (2020) transform 2D points into 2D lines. By extension, Lee et al. (2023) lift pairs of points to lines and Moon et al. (2024), relying on a set of spatial anchors, obtain a set of lines with non uniform direction distribution to make the inversion more challenging. Another strategy consists in swapping coordinates between random pairs of points (Pan et al., 2023) or performing partial pose estimation against distributed partials maps (Geppert et al., 2022). Such obfuscation methods have also been used in the context of SLAM (Shibuya et al., 2020) and Structure-from-Motion (SfM) (Geppert et al., 2020; 2021). These schemes aim to prevent inversion attacks by hiding the 2D positions, thus avoiding the applicability of the attacks. However, as shown by Chelani et al. (2021; 2025), the point positions may be recovered by identifying point neighborhoods, *e.g.*, from co-occuring descriptors, thus again enabling inversion attacks. Therefore, more recent privacy-preserving VL methods focus on obfuscating the descriptors and representations rather than the geometry. Dusmanu et al. (2021); Ng et al. (2022) and Pittaluga & Zhuang (2023) lift descriptors to affine subspaces or to an obfuscated manifold in order to prevent direct inversion. Zhou et al. (2022) and Wang et al. (2024) perform matching based solely on geometry, eliminating the need for image-based descriptors at the cost of reduced localization accuracy. In an orthogonal direction, Pietrantoni et al. (2023; 2025a) demonstrates that replacing high-dimensional descriptors with segmentation labels containing less information effectively prevents inversion attacks. In this work, we propose a shared inversion framework and evaluation protocol to assess the privacy level of most of these methods.

**Conditional Diffusion Models.** Diffusion models have emerged as a powerful class of generative models, enabling high-fidelity synthesis of complex data distributions by gradually adding noise to data and learning to reverse the diffusion process to generate samples (Ho et al., 2020; Dhariwal & Nichol, 2021). Subsequent works have focused on improving sampling through methods like DDIM Song et al. (2020) and DPM Lu et al. (2022), developing alternative formulation such as a continuous-time framework Feng et al. (2023) or improving visual fidelity with models using techniques like CLIP-based text encoders and denoising in a latent space Rombach et al. (2022); Saharia et al. (2022). Conditional diffusion models Ho et al. (2020); Rombach et al. (2022) convert conditioning signals into coherent and semantically meaningful data samples. Local conditioning signals such as depth, edges or human pose aim controlling the structure of the generated samples while signals such as text aim at controlling the style of the generated samples. Many structures have been introduced to efficiently incorporate the conditioning Zhang et al. (2023); Ye et al. (2023); Mou et al. (2024). In this work, we finetune a diffusion model Rombach et al. (2022) jointly conditioned on different representations and aim to measure the level of private data still contained in the representations.

## 3 DIFFUSION-BASED PRIVACY ATTACK

We first describe our privacy attack, which consists of a diffusion model trained to capture image distributions conditioned on a variety of dense or sparse image representations. Then, we introduce our multi-modal low rank adaptation (LoRA) finetuning scheme that allows us to further enhance training efficiency and to leverage the representational priors of large pretrained models. An overview of our pipeline is provided in Fig. 1.

**Conditioning diffusion for inversion.** Our privacy attack involves reconstructing an original image from its privacy-preserving representation using a diffusion model, conditioned on this representation. Diffusion models Ho et al. (2020); Dhariwal & Nichol (2021) are generative models characterized by two key processes: a forward process, where noise is iteratively added to the training

Figure 1: Overview of our privacy attack pipeline. Given a privacy preserving representation (fine-grained segmentation or a set of keypoints, *e.g.*, obtained from de-obfuscation Chelani et al. (2025), and descriptors), the network progressively denoises the noisy input tensor in a latent space before decoding it to the image associated with the representation. The network is conditioned locally on the privacy preserving representation through concatenation and globally through cross attention by encoding the set of keypoints or mask centroids with a GNN. LoRA modulation further allows for the inversion of different types of segmentation or descriptors with a single model.

sample, and a backward process, where a network learns to denoise the noisy sample by iteratively predicting the added noise. The forward process consists in adding Gaussian noise to a sample $x_0$ drawn from the true distribution q, gradually transforming it into a noise-only distribution. The probability of transitioning is described by $q(x_t|x_{t-1}) = N(x_t, \sqrt{1-\beta_t}x_{t-1}, \beta_t I)$, where $\beta_t$ is a timestep-dependent noise level. The reverse process aims at predicting $x_t$ from $x_{t-1}$ and is formulated as $p_\theta(x_{t-1}, x_t) = N(x_{t-1}, \mu_\theta(x_t, t), \Sigma_\theta(x_t, t))$, where $\mu_\theta$ and $\Sigma_\theta$ are parametrized by a neural network called denoising network. In practice, the denoising network predicts the noise and the training objective consists in minimizing the KL divergence between between the true (unknown) and approximated reverse process distributions. This objective can be simplified by minimizing the difference between predicted noise and the noise added during the forward process

$$L = E_{x_0, \epsilon \; N(0, \mathbf{I})} \|\epsilon - \epsilon_\theta(x_t, t)\| \ ,$$

with $x_t = \sqrt{\alpha_t}x_0 + \sqrt{1 + \alpha_t}\epsilon$, where $\alpha_t = \prod_{s=1}^t (1 - \beta_s)$ is a closed-form reparameterization used to simplify training objectives.

In this work, we elect to condition the reverse process on a privacy-preserving representation $c_{pp}$. The reverse process therefore becomes

$$p_\theta(x_{t-1}|x_t, c_{pp}) = N(x_{t-1}, \mu_\theta(x_t, t, c_{pp}), \Sigma_\theta(x_t, t, c_{pp})) \ ,$$

and the resulting training objective for the inversion network is $L = E_{x_0, \epsilon \sim N(0, \mathbf{I})} \|\epsilon - \epsilon_\theta(x_t, c, t)\|$. For each time step $t$ of the schedule, the conditioning is performed locally on the pixel-level by concatenating, on the channel dimension, the privacy-preserving representation to the noisy sample $x_t$. The resulting concatenated tensor is fed to the denoising network which extracts the information contained within the privacy-preserving representation to progressively denoise the sample.

To strengthen the conditioning signal and ensure maximal fidelity in the reverse process, we furthermore add a global conditioning through cross-attention layers in the denoising network. Given a set of $N$ keypoint locations or $N$ centroid center locations, we divide the image space in $S$ subwindows. Within each subwindow $s$, we construct a sparse graph using the keypoints/centers, and apply a graph attention network to compress structural information, yielding an embedding $g_s$ that captures the compositionality of the privacy-preserving representation. The $S$ embeddings are concatenated and passed to the cross-attention layers of the denoising network.

**Multi-modal low rank adaptation.** To leverage the semantic and structural priors contained within a pretrained text-to-image diffusion foundation model Rombach et al. (2022) and to compensate for the scarcity of training data associated with privacy-preserving representations, we use Low-Rank Adapdation (LoRA) Hu et al. (2022). LoRA is a parameter-efficient fine-tuning technique based on the hypothesis that weight updates during fine-tuning have low intrinsic dimensionality, enabling the adaptation of models with a large number of parameters. Formally, weights matrices of the original model are frozen and a new set of low-rank matrices are instead introduced. A weight matrix $\mathbf{W} \in R^{dk}$ belonging to the original model is adapted as follows: $\mathbf{W}' = \mathbf{W} + \frac{\alpha}{r}\mathbf{BA}$, where $\mathbf{A} \in R^{dr}$ and $\mathbf{B} \in R^{rk}$ are the low rank matrices, $r << min(d, k)$ the target rank, and $\alpha$ controls

the adaptation strength. The adaptation happens on a per layer basis, as such two low rank matrices are learnt and stored per layer.

To further improve both the robustness and the efficiency of our inversion model, instead of training a set of LoRA weights for each representation type that we want to invert, we train a single set of LoRA weights per group of similar representations and modulate the low-rank adaptation accordingly. We hypothesize that representations within the same group (for example sparse descriptors or fine grained segmentations) are sufficiently similar to enable the geometric and semantic information required for inversion to be captured by shared low-rank matrices. This shared representation leverages similarities between representations which could yield synergies while the signal is simply modulated to account for the differences between individual representations. Inspired by Stracke et al. (2024), we perform modulation within the LoRA low-rank subspace to enable regularization and prioritize critical details essential for inversion. Concretely, given a group of $C$ representations $\{c_i\}_{i=1}^{C}$, a learnable embedding $h_c^l$ is assigned to each representation $c$ for every layer $l$, which serves as input to two small MLPs predicting a scale $s = f_s^l(h_c^l)$ and a bias $b = f_b^l(h_c^l)$. Given an input $x_c$ conditioned on $c$ and the weight matrix $\mathbf{W}$ associated to a layer $l$, the modulation for the layer is then effectively described by

$$\mathbf{W}'x_c = \mathbf{W}x_c + \tfrac{\alpha}{r}(\mathbf{B} + (f_s^l(h_c) * \mathbf{A}x_c + f_b^l(h_c)))\ ,$$

yielding an efficient model that can invert multiple representations with limited overhead and efficient training.

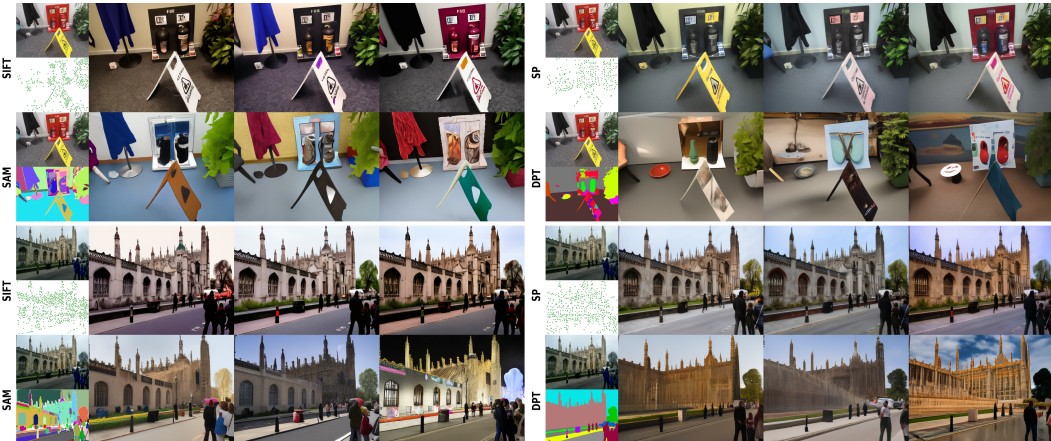

Figure 2: Reconstruction output for different seed and similar conditioning. Visualization of SIFT/SuperPoint/DPT/SAM2 conditioning. The ground truth image is displayed on the top left and the conditioning is displayed on the bottom left. As opposed to segmentations, SIFT/Superpoint display low variability under different seeds.

## 4 PRIVACY EVALUATION PROTOCOL

To assess the privacy level of a given obfuscated representation, we first try recovering the associated image with our diffusion-based model and then evaluate the quality and semantic content of this image with various perceptual and semantic metrics. The inversion model extracts information contained in the input representation by mapping the latent representation into an interpretable RGB space, enabling both visual and quantitative comparison with the original image content. The less privacy-preserving the representation is, the more information it contains and the more faithful the reconstructed image will be. Conversely, higher privacy-preservation yields less detailed or less faithful image reconstructions.

Our models are trained on a combination of two large-scale datasets, MegaDepth Li & Snavely (2018) and ScanNet-v2 Dai et al. (2017), which account for a total of two million training images covering a wide variety of outdoor and indoor scenes. Two inversion models are trained: one for sparse descriptors (using SIFT Lowe (1999), Superpoint DeTone et al. (2018), and Xfeat Potje et al. (2024) features) and one for segmentations (using segmentations generated using SegLoc Pietrantoni et al. (2023), GSFF Pietrantoni et al. (2025a), DPT Ranftl et al. (2021), and SAM2 Ravi et al.

| | | Conditioning | LPIPS ↑ | FID ↑ | SSIM ↓ | RD(SSIM) ↓ | RD(LPIPS) ↑ | SCR ↓ |
|---|---|---|---|---|---|---|---|---|
| Cambridge | SDesc. | SIFT | 0.29 | 78 | 0.48 | 0.56 | 0.18 | 0.72 |
| | | XFeat | 0.26 | 73 | 0.58 | 0.63 | 0.12 | 0.74 |
| | | SuperPoint | 0.19 | 60 | 0.57 | 0.57 | 0.17 | 0.71 |
| | Seg. | SAM2 | 0.48 | 118 | 0.32 | 0.38 | 0.41 | 0.63 |
| | | DPT ADE20k | **0.58** | **146** | **0.29** | **0.31** | **0.53** | **0.60** |
| 7-Scenes | SDesc. | SIFT | 0.36 | 146 | 0.62 | 0.69 | 0.21 | 0.74 |
| | | XFeat | 0.21 | 105 | 0.75 | 0.84 | 0.10 | 0.77 |
| | | SuperPoint | 0.22 | 108 | 0.73 | 0.77 | 0.12 | 0.78 |
| | Seg. | SAM2 | 0.50 | 194 | 0.49 | 0.52 | 0.38 | 0.63 |
| | | DPT ADE20k | **0.60** | **240** | **0.46** | **0.48** | **0.50** | **0.57** |

Table 1: Inversion results on Cambridge Landmarks and 7-Scenes datasets comparing conditioning on local features *versus* segmentations. Higher FID,LPIPS,RD(LPIPS) as well as lower SSIM, RD(SSIM),SCR indicates lower quality uncertain reconstruction implying higher degree of privacy.

(2024)), both with a LoRA rank of 64. More details are provided in supp. material. After training, we apply our inversion models on the two most commonly used real-world datasets in visual localization works, *i.e.*, 7Scenes (Shotton et al., 2013) and Cambridge Landmarks (Kendall et al., 2015). Importantly, none of these evaluation datasets were seen by the inversion model during training. To measure privacy, we employ a range of evaluation metrics:

- **Perceptual similarity** is measured via LPIPS (Zhang et al., 2018) and FID (Heusel et al., 2017). LPIPS (Learned Perceptual Image Patch Similarity) relies on a deep network that was trained to align closely with human visual judgment. FID (Fréchet Inception Distance) quantifies the realism and diversity of the reconstructed images. These metrics provide a broad overview of the perceived visual fidelity.

- **Structural similarity** is measured via SSIM (Structural Similarity Index Measure) (Wang et al., 2004), which measures image degradation as a perceived change in structural information, while also incorporating important perceptual phenomena, including both luminance masking and contrast masking terms.

- **Reconstructed Diversity (RD)** is a new measure we propose in this paper. Given a set of reconstructed images obtained with the same conditioning representation but different random seeds for input, we compute pairwise SSIM and LPIPS for this image set. High pairwise SSIM/low pairwise LPIPS between pairs suggests that the conditioning input tightly governs the reconstruction, whereas low pairwise SSIM/high pairwise LPIPS implies the diffusion model had to infer or 'hallucinate' details due to insufficient guidance by the representation, which indicate that the conditioning representation is more privacy-preserving.

- **Semantic Content Recovery** is a metric (Pietrantoni et al., 2025b) aimed to evaluate the amount of semantic information (*i.e.*, interpretable concepts for a human) contained in the reconstructed image by querying a visual-language model (VLM) (Liu et al., 2024) which describes the content of the images in fine details. We encode these descriptions with KeyBert (Grootendorst, 2020) and compute cosine similarity between their embeddings to evaluate the semantic content alignment between the two images.

## 5 EXPERIMENTAL RESULTS

In this section, we first perform a more general preliminary study, where we assess the image content recovered by the proposed diffusion-based inversion model conditioned on segmentations and sparse descriptors. Building upon this, we then compare the privacy preservation level of several privacy-preserving visual localization (PPVL) methods. Finally, we demonstrate the superiority of our diffusion-based approach over the feed-forward convolutional privacy attack (FFConv) proposed by Pittaluga et al. (2019) and provide further ablations.

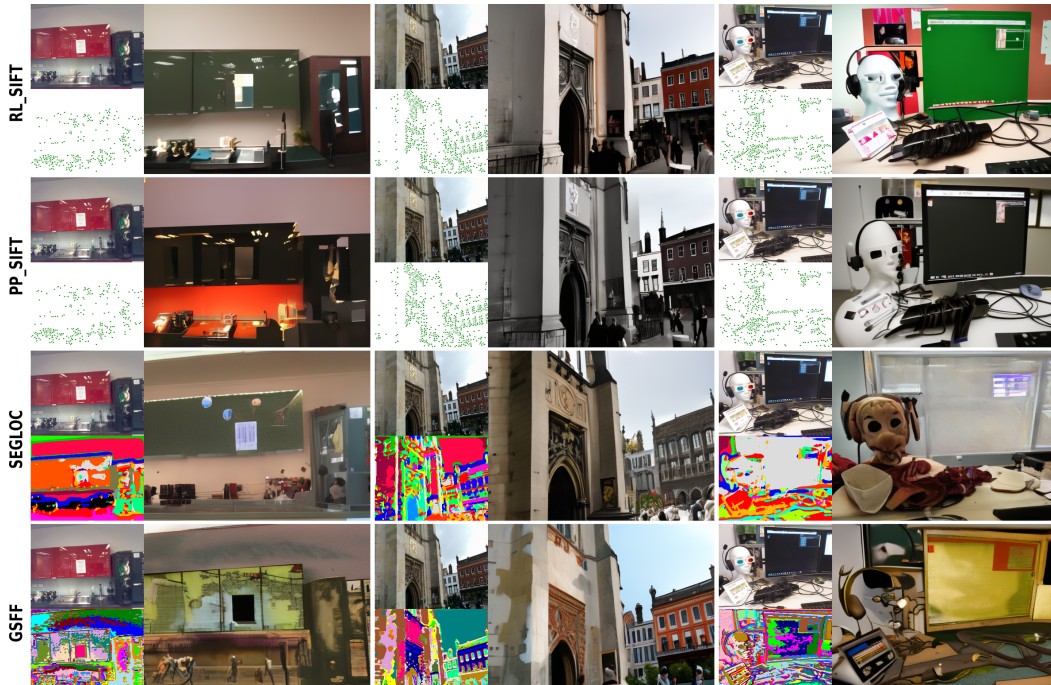

Figure 3: Reconstructions from various privacy-preserving visual localization schemes: SIFT Random lines (RL SIFT) Speciale et al. (2019b), SIFT permutation (PP SIFT) Pan et al. (2023), Segloc Pietrantoni et al. (2023), and GSFF Pietrantoni et al. (2025a). The ground truth image is displayed on the top left and the conditioning is displayed on the bottom left. All methods capture the coarse structure of the image, however segmentations better obfuscate small details/textures.

## 5.1 INVERSION RESULTS OF LOCAL FEATURES *versus* SEGMENTATIONS

We start this experimental section with a more general study aimed at evaluating the amount of information encoded in segmentations and local descriptors by inverting them with our model. Thus, we evaluate the faithfulness of images reconstructed from semantic segmentations generated by DPT (Ranftl et al., 2021) (trained on ADE20k (Zhou et al., 2019)), from segmentation masks obtained with SAM2 Ravi et al. (2024), and compare them to images reconstructed from sparse descriptors SIFT (Lowe, 1999), SuperPoint (DeTone et al., 2018), and XFeat Potje et al. (2024). Results for both the 7Scenes dataset and the Cambridge Landmarks dataset are reported in Table 1 and visual examples are shown in Fig. 2. We average the metrics first over images in the scenes, then report the average over the scenes in the tables. Full tables with per scene results can be found in the Supplementary.

**Sparse local descriptors encode sufficient information to enable high-fidelity image reconstruction.** This is shown by all metrics in Table 1, demonstrating that even a sparse set of local descriptors (between 500 and 1000 keypoints per image) retains sufficient data to enable high-fidelity image reconstruction. Low perceptual differences (LPIPS, FID) and high structural similarity (SSIM) suggest that reconstructions are perceptually and statistically close to original images. Further, high SCR scores suggest that the VLM model was able to recognize the semantic content in these images well.

**Learnt *versus* handcrafted features.** Learned descriptors (XFeat, SuperPoint) encode more information than handcrafted descriptors such as SIFT. Note that the difference in information is not necessarily due to descriptor dimensionality (XFeat has 64 channels, SuperPoint 256, and SIFT 128) but rather due to the training objective and higher representational objective of deep networks for the learned descriptors compared to the gradient binning in SIFT.

**Dense segmentations are harder to invert.** Segmentations are inherently ambiguous due to the wide range of potential visual content that a single mask can encode, *i.e.*, there are multiple visually dissimilar images that can lead to the same segmentation. This ambiguity is reflected in the Recon-

| | | Conditioning | LPIPS ↑ | FID ↑ | SSIM ↓ | RD(SSIM) ↓ | RD(LPIPS) ↑ | SCR ↓ |
|---|---|---|---|---|---|---|---|---|
| Cambridge | SDObf. | Coord. Perm. (SIFT) | 0.35 | 85 | 0.41 | 0.50 | 0.23 | 0.71 |
| | | Random Lines (SIFT) | 0.33 | 81 | 0.41 | 0.51 | 0.22 | 0.73 |
| | | Coord. Perm. (SuperPoint) | 0.29 | 66 | 0.43 | 0.57 | 0.17 | 0.74 |
| | | Random Lines (SuperPoint) | 0.28 | 67 | 0.43 | 0.57 | 0.17 | 0.74 |
| | Seg. | SegLoc | 0.40 | 92 | 0.41 | 0.47 | 0.32 | 0.64 |
| | | GSFF | **0.43** | **98** | **0.33** | **0.39** | **0.36** | **0.63** |
| 7-Scenes | SDObf. | Coord. Perm. (SIFT) | 0.49 | 187 | 0.56 | 0.67 | 0.33 | 0.62 |
| | | Random Lines (SIFT) | 0.46 | 170 | 0.57 | 0.66 | 0.31 | 0.64 |
| | | Coord. Perm. (SuperPoint) | 0.40 | 184 | 0.60 | 0.72 | 0.22 | 0.68 |
| | | Random Lines (SuperPoint) | 0.37 | 168 | 0.62 | 0.72 | 0.21 | 0.69 |
| | Seg. | SegLoc | 0.48 | 186 | 0.54 | 0.56 | 0.34 | 0.60 |
| | | GSFF | **0.50** | **195** | **0.38** | **0.42** | **0.39** | **0.59** |

Table 2: Inversion results on the Cambridge Landmarks and 7-Scenes datasets comparing inversion results conditioned on PPVL representations. Higher FID,LPIPS,RD(LPIPS) as well as lower SSIM, RD(SSIM),SCR indicates lower quality uncertain reconstruction implying higher degree of privacy.

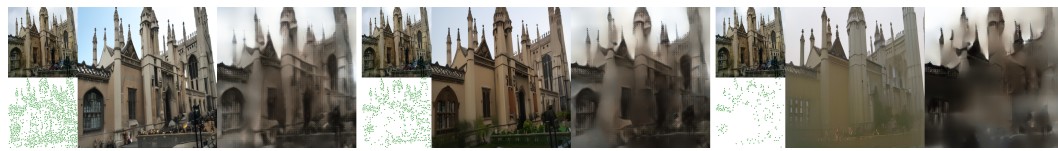

Figure 4: Comparison of reconstructions between our inversion model (left) and a feed-forward convolutional inversion model (right) for different level of keypoint sparsity (from left to right: denser to sparser).

structed Diversity (RD) scores on both datasets: when sampling different initial noises, the inversion model conditioned on segmentations produces wildly different images for a similar conditioning signal, leading to high pairwise LPIPS and low pairwise SSIM. In contrast, sparse descriptors produce stable (up to (slight) variations in color) and accurate outputs (see Fig. 2).

**Finer segmentation granularity helps inversion.** SAM2, despite having per-image masks whose concepts do not generalize across images, yields better inversion metrics than DPT which segments semantic concepts. As shown in Fig. 2, SAM2 masks are much more fine grained than the broad semantic masks created by DPT, which in turn results in much more discriminative segmentations proving less obfuscation than broader and less numerous masks.

### 5.2 INVERSION RESULTS FROM PPVL REPRESENTATIONS

Reducing the encoded information (*e.g.*, fewer descriptors or fewer segmentation masks) inherently increases privacy, but a minimum level of discriminative information is still necessary for accurate visual localization. In this section, we thus compare the vulnerability of several PPVL methods and extract the representations with the hyperparameters that provide the best privacy-discriminativeness trade-off. We compare four PPVL methods that represent the state-of-the-art in terms of localization accuracy: we consider two sparse feature-based obfuscation methods, Random Lines Speciale et al. (2019a) and Coordinate Permutation Pan et al. (2023), and two segmentation-based methods, Segloc (Pietrantoni et al., 2023) and GSFF (Pietrantoni et al., 2025a). For the geometric obfuscation methods, we first de-obfuscate the keypoints using (Chelani et al., 2025), then use the recovered keypoints in combination with the sparse descriptors to condition the inversion process. The average numbers of keypoints per image per scene are reported in the supplementary material. For the segmentation methods, we use the dense segmentation of the query image directly as the inversion model's conditioning signal. Results for both the 7Scenes and Cambridge Landmarks datasets are reported in Table 2, evaluation protocol and metrics are the same as in the last section. In Fig. 3 we qualitatively compare a few images reconstructed from different PPVL representations.

**Segmentation-based PPVL are more robust against privacy attack.** The higher vulnerability of local features observed in Section 5.1 also holds when we evaluate the geometric obfuscation PPVL representations. For both Random Lines and Coordinate Permutation, perceptual and semantic re-

| | SIFT | | | Segloc | | |
|---|---|---|---|---|---|---|
| | FFConv | Ours no GC | Ours | FFConv | Ours no GC | Ours |
| CL | 0.450 / 0.438 / 164.0 | 0.460 / 0.308 / 78.3 | 0.492 / 0.283 / 74.3 | 0.370 / 0.508 / 229.3 | 0.395 / 0.407 / 94.8 | 0.412 / 0.392 / 92.0 |

| | SIFT High Sparsity | | SIFT Medium Sparsity | | SIFT Low Sparsity | |
|---|---|---|---|---|---|---|
| | FFConv | Ours | FFConv | Ours | FFConv | Ours |
| CL | 0.382 / 0.588 / 247.5 | 0.405 / 0.485 / 130.0 | 0.427 / 0.502 / 197.5 | 0.455 / 0.360 / 96.8 | 0.450 / 0.438 / 164.0 | 0.492 / 0.283 / 74.3 |

Table 3: Inversion experiments on Cambridge Landmarks (CL) for SIFT descriptors and SegLoc segmentations. Comparison of Feed Forward convolutional approach (FFConv) *vs.* our diffusion based approach in term of reconstruction metrics (SSIM($\uparrow$), LPIPS ($\downarrow$), FID ($\downarrow$)).

covery metric in Table 2 indicate that the inversion model is able to recover more image content from the de-obfuscated keypoints than from SegLoc and GSFF. The variability analysis also clearly indicates that segmentations result in more uncertainty during the inversion. Finally, GSFF is the most privacy-preserving visual localization method followed by SegLoc

**Robustness of the inversion model to noisy keypoint locations.** Even though the keypoint recovery process for permutation and random lines results in 5 to 10 pixels of median error in terms of recovered point positions (Chelani et al., 2025), this level of noise in the recovered keypoint positions still allows for the inversion model to reconstruct images with satisfying fidelity which, in turn, results in a lower degree of privacy for geometric obfuscation methods. Coordinate permutation generally results in a slightly higher degree of privacy than random lines.

**Segmentations privacy-discriminativeness tradeoff.** Both GFSS and SegLoc segmentation classes are learnt in a self-supervised way which inherently optimizes for the discriminativess of the segmentation (as it is crucial characteristic for the downsteam localization task). This training regime therefore results in a slightly lower degree of privacy than other segmentations such as DPT/SAM2. We further observe that GFSS segmentations leak less amount of sensitive information than SegLoc, which may be caused by the generalizable nature of Segloc in combination with its higher number of classes compared to the per-scene segmentations of GSFF.

### 5.3 GLOBAL CONDITIONING AND COMPARISON TO CONVOLUTIONAL INVERSION MODEL

We compare our diffusion-based inversion model to a feed forward convolutional inversion model (FFConv). FFConv takes as input the same local conditioning as our diffusion-based model and is trained to reconstruct images with a L2 and perceptual loss. One FFConv model is trained on SIFT and another one on SegLoc. Results for Cambridge Landmarks are reported in Tab. 3 with three levels of sparsity for SIFT keypoints (S1: avg. 2k kpts/img, S2: avg. 1k kpts/img, S3: avg. 500kpts/img). The feed-forward approach is able to approximately recover the structure of the images (as illustrated by the slight drop of SSIM compared to our privacy attack) but mostly fails at recovering the content and style of the images. The style and content are the most critical aspects when it comes to privacy assessment, thus making our privacy attack a better tool for evaluating privacy. Our diffusion approach also shows higher robustness under a higher level of sparsity. Visual examples are shown in Fig. 4. Ablating the global structure conditioning (No GC) also leads to lower reconstruction metrics underlining its importance in guiding the denoising process.

## 6 CONCLUSION

Motivated by the increasing importance of privacy in visual localization, this paper attempted to compare different privacy-preserving methods within a new common comprehensive evaluation framework. We introduced a novel diffusion-based privacy attack in which a diffusion model, conditioned locally and globally on privacy-preserving representations, learns to denoise and reconstruct images. Evaluating the quality of these reconstructed images with perceptual/diversity/semantic metrics allows us to quantify the amount of information encoded in the associated privacy-preserving representation. This subsequently acts as a proxy to measure the degree of privacy of the representation. We showed that segmentations provide a better degree of privacy than geometric obfuscation methods as the latter rely on deep sparse descriptors encoding sensitive information in fine details. In contrast, segmentation masks are inherently more ambiguous and encode less information although this can be controlled with the granularity and the degree of semantic content in the concepts captured by such classes. We hope that this work will open the way for more considerations around privacy when designing and learning representations for visual localization.

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
