# Supplementary material: Vulnerability of Privacy-Preserving Visual Localization against Diffusion-based Attacks

In this supplementary material we first provide additional training and implementation details. We then present per-scene inversion metrics for all the studied baselines. Finally, we display additional visualizations of sparse descriptors and segmentations inversions in Fig. 1 and privacy-preserving representations inversions in Fig. 2.

## A  Additional training and implementaiton details

In addition to the LoRA layers, we also optimize a distinct input convolutional layer per type of representation, enabling the conditioning of groups of representation with different dimensions. The LoRA rank is set to 64 providing a good compromise between representational power and efficiency. Increasing the rank would allow for better inversions at the cost of increased training / inference time and higher storage requirements. The $\alpha$ LoRA adaptation controls parameter is set to 12. Each model is trained on 4 Nvidia V100 32Gb GPUs with a learning rate of 1e-4 for 5 days. Images are resized to 512x512 both for training and inference. We use a DDPM scheduler (Lu et al., 2022) with 1000 steps during training and 25 steps during inference. Conditional free guidance is set to 1.5 for both sparse descriptors and segmentation models. During training, the global structural conditioning is zeroed out 10 percent of the time. Random gaussian noise with a std of 4 pixels is applied to the keypoint locations during training. We use the cv2 implementation of SIFT as well as the available public codebase/checkpoint for SuperPoint. Average number of keypoints per image utilized in the main experiments are reported in Tab. 1. Both methods extract keypoints and descriptors from grayscale images. For SegLoc we use a single trained encoder that extracts hierarchical segmentations from images. We keep the finest segmentation with 100 classes as it is the most informative. GSFF is a per-scene method, therefore we use a different encoder trained for each scene. The encoder extracts 34 classes segmentations from images. Public checkpoints and implementations are used SAM2 and DPT Ade20k. Each class is mapped to a 3 dimensional RGB color.

For sparse descriptors, SIFT (Lowe, 1999), SuperPoint (DeTone et al., 2018), XFeat (Potje et al., 2024) keypoints and descriptors are extracted, then embedded into a sparse tensor aligned with the latent space. Descriptors are inserted into the channel dimension at keypoint locations, forming the conditioning tensor. For segmentations, labels are extracted from the image-based encoders of DPT semantic segmenter (Ranftl et al., 2021), Segloc (Pietrantoni et al., 2023), SAM2 (Ravi et al., 2024) and GSFF (Pietrantoni et al., 2025). Labels are then mapped to 3 dimensional RGB colors, and downsampled to match the latent space's spatial dimensions. The resulting tensor is used as local conditioning through concatenation. In both cases, batches are sampled to ensure a single type of sparse descriptor or a single type of segmentation per batch.

The sparse feature based obfuscation methods take as input a set of SIFT or SuperPoint keypoints+descriptors and obfuscate the keypoints positions by either coordinate permutations (Pan et al., 2023) or by replacing keypoints by random lines (Speciale et al., 2019). The descriptors are kept intact and used to compute matches. The pose is then estimated with minimal solvers adapted for obfuscated representations. Segloc (Pietrantoni et al., 2023) and GSFF (Pietrantoni et al., 2025) solely rely on segmentation for pose estimation, poses are estimated through optimization via segmentation alignment between a query segmentation and segmentation lifted in a 3D scene representation.

|  | Chess | Fire | Heads | Office | Pumpkin | Redkitchen | Stairs |
|---|---|---|---|---|---|---|---|
| SIFT | 770 | 1175 | 448 | 387 | 451 | 632 | 414 |
| SUPERPOINT | 740 | 672 | 605 | 552 | 679 | 768 | 463 |
|  | KingsCollege | OldHospital | ShopFacade | StMarysChurch |  |  |  |
| SIFT | 1601 | 2680 | 1631 | 1472 |  |  |  |
| SUPERPOINT | 1311 | 1692 | 1411 | 1163 |  |  |  |

Table 1: Average number of keypoints per image per scene.

# B    PER SCENE METRICS

Per scene privacy metrics for Segmentations (Seg.), Geometric Obfuscation methods (SDObf.) and Sparse Descriptors (SDesc.) methods are presented in Tab. 2 for the 7-Scenes dataset and in Tab. 3 for the Cambridge Landmarks dataset. Furthermore, per-scenes results for the ablation study are presented in Tab. 4.

| | Input | Chess | Fire | Heads | Office | Pumpkin | Redkitchen | Stairs |
|---|---|---|---|---|---|---|---|---|
| | | SSIM (↓), LPIPS (↑), FID (↑) | | | | | | |
| Seg. | DPT ADE20k | 0.43/0.57/244 | 0.42/0.58/228 | 0.33/0.67/403 | 0.51/0.55/129 | 0.53/0.58/213 | 0.46/0.61/190 | 0.54/0.59/277 |
| Seg. | SAM2 | 0.46/0.49/189 | 0.44/0.51/211 | 0.45/0.52/268 | 0.55/0.47/114 | 0.57/0.49/164 | 0.47/0.53/161 | 0.51/0.51/256 |
| Seg. | GSFF | 0.42/0.47/173 | 0.33/0.48/214 | 0.33/0.55/287 | 0.38/0.53/161 | 0.46/0.51/174 | 0.33/0.51/149 | 0.43/0.46/210 |
| Seg. | SegLoc | 0.51/0.59/186 | 0.49/0.44/204 | 0.47/0.51/293 | 0.59/0.44/106 | 0.61/0.46/164 | 0.51/0.50/153 | 0.62/0.41/190 |
| SDObf. | SIFT Permutation | 0.54/0.47/190 | 0.46/0.43/200 | 0.55/0.54/221 | 0.59/0.51/124 | 0.64/0.50/207 | 0.54/0.44/135 | 0.63/0.53/230 |
| SDObf. | SIFT Random Lines | 0.55/0.45/179 | 0.46/0.40/188 | 0.58/0.50/192 | 0.60/0.48/112 | 0.63/0.48/186 | 0.54/0.45/131 | 0.61/0.49/203 |
| SDObf. | SuperPoint Permutation | 0.58/0.38/212 | 0.50/0.41/192 | 0.61/0.39/198 | 0.66/0.37/116 | 0.67/0.40/182 | 0.59/0.37/148 | 0.62/0.47/240 |
| SDObf. | SuperPoint Random Lines | 0.60/0.35/195 | 0.51/0.38/175 | 0.63/0.36/188 | 0.67/0.34/99 | 0.68/0.37/169 | 0.60/0.35/128 | 0.63/0.43/220 |
| SDesc. | SIFT | 0.63/0.32/146 | 0.49/0.33/166 | 0.63/0.36/140 | 0.64/0.37/93 | 0.67/0.39/171 | 0.59/0.36/145 | 0.69/0.37/162 |
| SDesc. | XFeat | 0.77/0.20/112 | 0.61/0.25/141 | 0.79/0.19/112 | 0.79/0.19/72 | 0.79/0.22/101 | 0.72/0.23/103 | 0.80/0.21/96 |
| SDesc. | SuperPoint | 0.74/0.19/112 | 0.59/0.25/142 | 0.77/0.19/111 | 0.78/0.20/68 | 0.76/0.22/110 | 0.71/0.21/100 | 0.76/0.25/110 |
| | | Mean Pairwise SSIM (↓), Mean Pairwise LPIPS (↓) | | | | | | |
| Seg. | DPT ADE20k | 0.44/0.49 | 0.53/0.44 | 0.34/0.56 | 0.49/0.46 | 0.51/0.50 | 0.49/0.50 | 0.54/0.52 |
| Seg. | SAM2 | 0.50/0.37 | 0.55/0.37 | 0.47/0.42 | 0.54/0.36 | 0.60/0.36 | 0.50/0.39 | 0.48/0.40 |
| Seg. | GSFF | 0.45/0.37 | 0.38/0.40 | 0.40/0.42 | 0.39/0.41 | 0.49/0.38 | 0.37/0.41 | 0.46/0.36 |
| Seg. | SegLoc | 0.49/0.37 | 0.59/0.29 | 0.48/0.40 | 0.57/0.34 | 0.62/0.36 | 0.50/0.40 | 0.64/0.25 |
| SDObf. | SIFT Permutation | 0.62/0.31 | 0.62/0.29 | 0.68/0.36 | 0.68/0.35 | 0.73/0.31 | 0.65/0.33 | 0.69/0.34 |
| SDObf. | SIFT Random Lines | 0.60/0.30 | 0.61/0.28 | 0.67/0.35 | 0.67/0.33 | 0.72/0.30 | 0.64/0.30 | 0.69/0.32 |
| SDObf. | SuperPoint Permutation | 0.70/0.22 | 0.66/0.24 | 0.73/0.22 | 0.75/0.21 | 0.77/0.21 | 0.71/0.21 | 0.75/0.25 |
| SDObf. | SuperPoint Random Lines | 0.70/0.20 | 0.66/0.22 | 0.74/0.20 | 0.75/0.20 | 0.76/0.21 | 0.72/0.19 | 0.74/0.23 |
| SDesc. | SIFT | 0.66/0.19 | 0.60/0.19 | 0.67/0.26 | 0.71/0.22 | 0.75/0.22 | 0.67/0.20 | 0.74/0.19 |
| SDesc. | XFeat | 0.82/0.11 | 0.73/0.12 | 0.86/0.09 | 0.87/0.10 | 0.88/0.10 | 0.82/0.11 | 0.87/0.08 |
| SDesc. | SuperPoint | 0.75/0.11 | 0.70/0.13 | 0.79/0.12 | 0.80/0.11 | 0.80/0.12 | 0.74/0.12 | 0.78/0.12 |
| | | Captions similarity (↓) | | | | | | |
| Seg. | DPT ADE20k | 0.61 | 0.52 | 0.46 | 0.68 | 0.56 | 0.63 | 0.56 |
| Seg. | SAM2 | 0.62 | 0.49 | 0.71 | 0.70 | 0.63 | 0.70 | 0.59 |
| Seg. | GSFF | 0.57 | 0.44 | 0.62 | 0.70 | 0.59 | 0.66 | 0.58 |
| Seg. | SegLoc | 0.60 | 0.46 | 0.50 | 0.72 | 0.61 | 0.66 | 0.63 |
| SDObf. | SIFT Permutation | 0.60 | 0.63 | 0.55 | 0.74 | 0.57 | 0.65 | 0.58 |
| SDObf. | SIFT Random Lines | 0.64 | 0.60 | 0.61 | 0.77 | 0.57 | 0.66 | 0.64 |
| SDObf. | SuperPoint Permutation | 0.69 | 0.62 | 0.79 | 0.75 | 0.59 | 0.67 | 0.65 |
| SDObf. | SuperPoint Random Lines | 0.68 | 0.61 | 0.77 | 0.81 | 0.62 | 0.66 | 0.67 |
| SDesc. | SIFT | 0.74 | 0.71 | 0.71 | 0.83 | 0.71 | 0.74 | 0.75 |
| SDesc. | XFeat | 0.76 | 0.67 | 0.79 | 0.85 | 0.75 | 0.76 | 0.79 |
| SDesc. | SuperPoint | 0.78 | 0.71 | 0.80 | 0.84 | 0.79 | 0.79 | 0.77 |

Table 2: Inversion experiments on 7-Scenes. Evaluation of the privacy level of Segmentations (Seg.), Geometric Obfuscation methods (SDObf.) and Sparse Descriptors (SDesc.) through different proxies: quality of reconstructed images (with SSIM / LPIPS / FID metrics), variability of the denoising process with the same conditioning input (Mean Pairwise SSIM / Mean Pairwise LPIPS computed over 10 images reconstructed from different seeds and same conditioning) and caption similarity between LLava description of the reconstructed images and ground truth images.

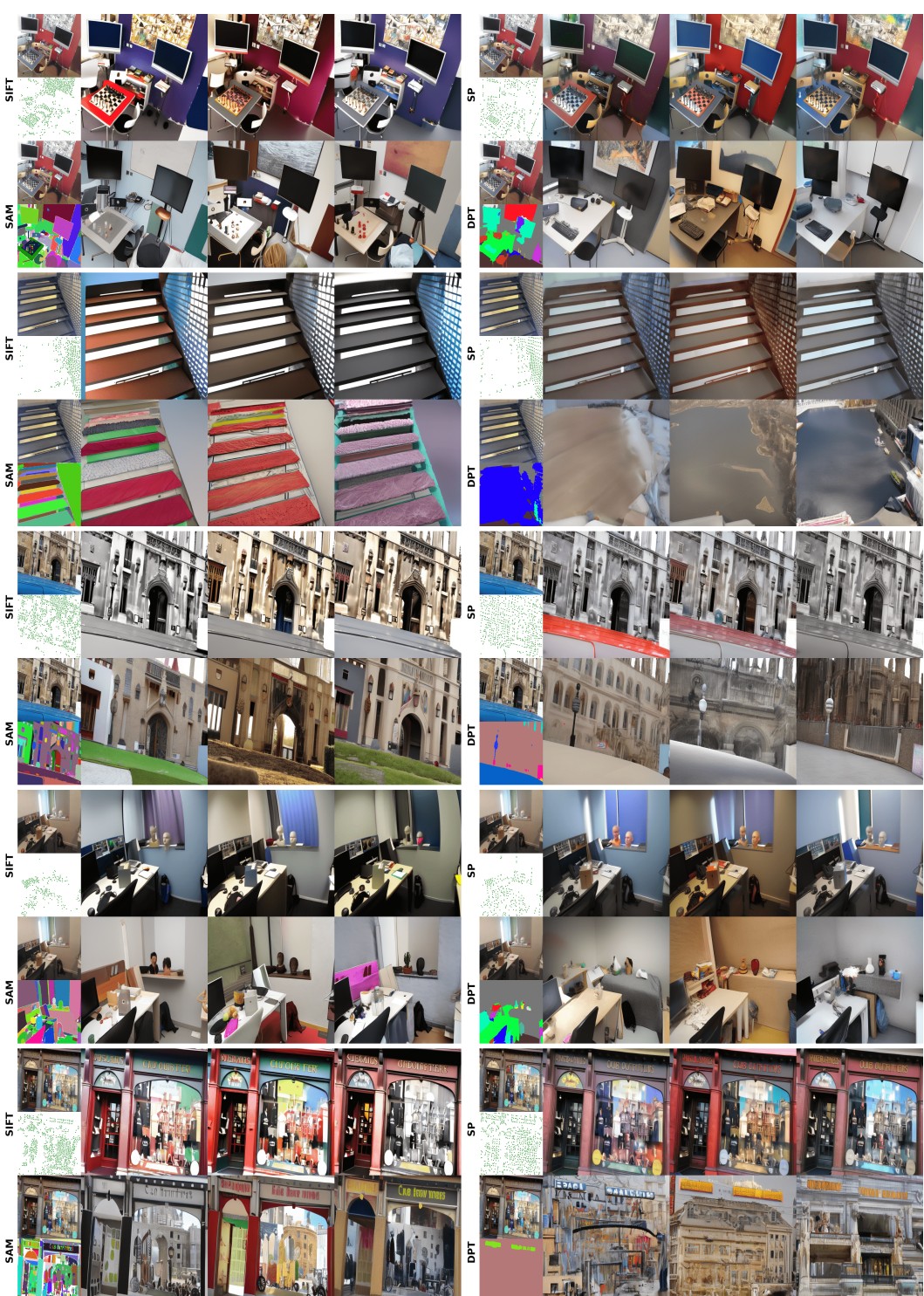

Figure 1: Reconstruction output for different seed and similar conditioning. Visualization of Sift/Superpoint/DPT/SAM2 conditioning. Gt image displayed on the top left and conditioning displayed on the bottom left.

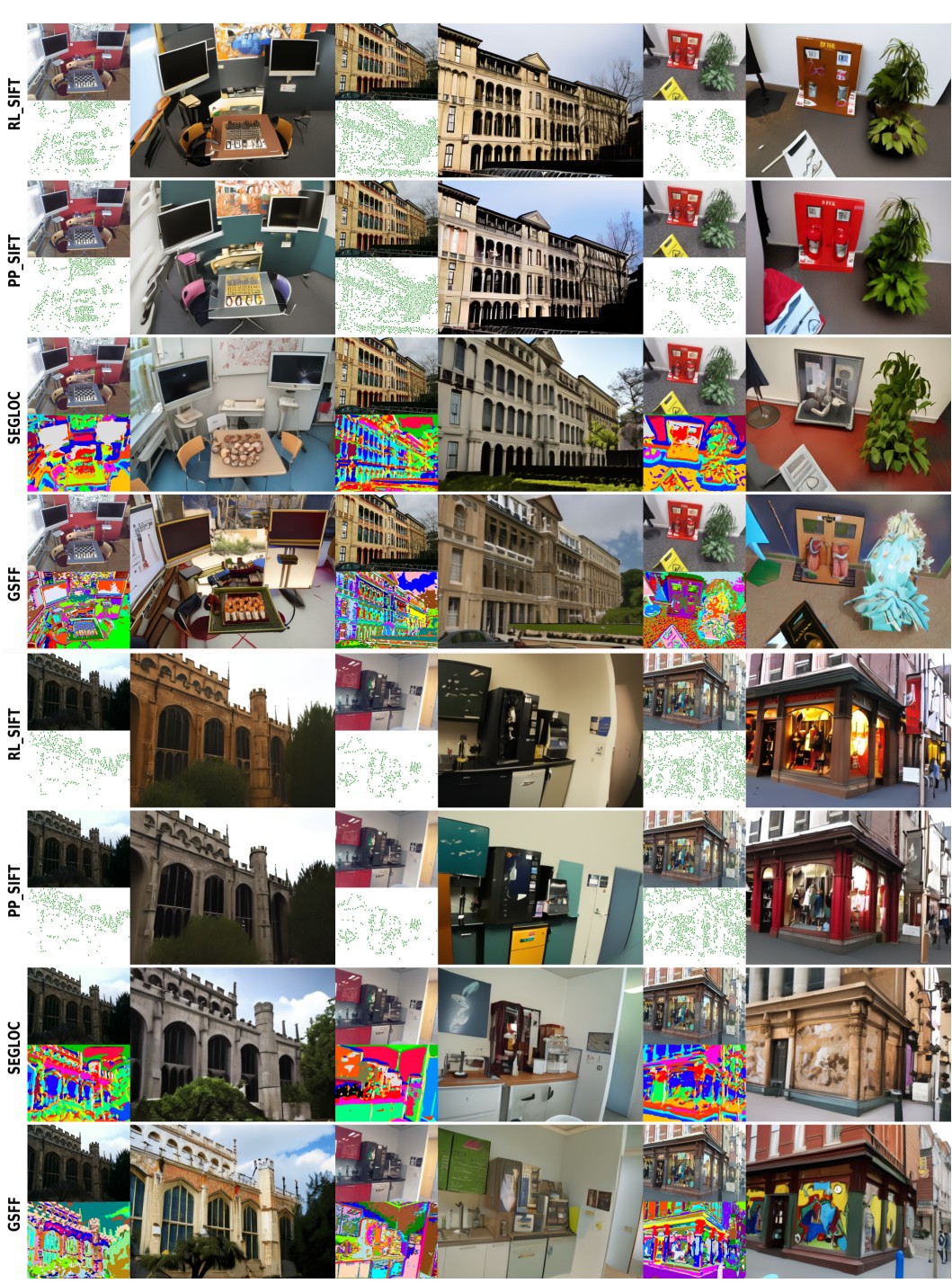

Figure 2: Reconstructions from various PPVL representations: SIFT Random lines (RL SIFT), SIFT permutation (PP SIFT), Segloc and GSFF. Gt image displayed on the top left and conditioning displayed on the bottom left.

| | Input | KingsCollege | OldHospital | ShopFacade | StMarysChurch |
|---|---|---|---|---|---|
| | | SSIM ($\downarrow$), LPIPS ($\uparrow$), FID ($\uparrow$) | | | |
| Seg. | DPT ADE20k | 0.37/0.51/73 | 0.20/0.62/177 | 0.24/0.63/239 | 0.34/0.56/95 |
| | SAM2 | 0.39/0.46/72 | 0.26/0.48/93 | 0.26/0.50/203 | 0.38/0.49/102 |
| | GSFF | 0.33/0.44/71 | 0.22/0.45/86 | 0.37/0.42/157 | 0.41/0.41/78 |
| | SegLoc | 0.47/0.37/55 | 0.35/0.36/72 | 0.38/0.44/168 | 0.45/0.40/73 |
| SDObf. | SIFT Permutation | 0.48/0.34/52 | 0.32/0.31/68 | 0.39/0.38/139 | 0.46/0.38/81 |
| | SIFT Random Lines | 0.48/0.32/48 | 0.32/0.30/64 | 0.39/0.36/138 | 0.45/0.35/74 |
| | SuperPoint Permutation | 0.47/0.29/35 | 0.35/0.29/69 | 0.43/0.30/107 | 0.47/0.28/54 |
| | SuperPoint Random Lines | 0.48/0.28/36 | 0.35/0.28/67 | 0.42/0.29/112 | 0.47/0.27/52 |
| SDesc. | SIFT | 0.56/0.28/52 | 0.42/0.24/60 | 0.47/0.30/121 | 0.52/0.31/64 |
| | XFeat | 0.62/0.25/55 | 0.53/0.24/72 | 0.57/0.27/107 | 0.59/0.26/56 |
| | SuperPoint | 0.62/0.18/37 | 0.49/0.19/59 | 0.55/0.21/98 | 0.60/0.19/47 |
| | | Mean Pairwise SSIM ($\downarrow$), Mean Pairwise LPIPS ($\uparrow$) | | | |
| Seg. | DPT ADE20k | 0.35/0.47 | 0.34/0.52 | 0.28/0.56 | 0.25/0.56 |
| | SAM2 | 0.42/0.40 | 0.37/0.37 | 0.34/0.43 | 0.38/0.44 |
| | GSFF | 0.36/0.37 | 0.37/0.33 | 0.44/0.36 | 0.38/0.38 |
| | SegLoc | 0.50/0.30 | 0.46/0.27 | 0.45/0.36 | 0.45/0.33 |
| SDObf. | SIFT Permutation | 0.56/0.20 | 0.48/0.22 | 0.51/0.25 | 0.449/0.23 |
| | SIFT Random Lines | 0.56/0.20 | 0.48/0.21 | 0.52/0.24 | 0.47/0.23 |
| | SuperPoint Permutation | 0.62/0.16 | 0.56/0.16 | 0.59/0.17 | 0.51/0.18 |
| | SuperPoint Random Lines | 0.61/0.16 | 0.57/0.15 | 0.59/0.17 | 0.51/0.18 |
| SDesc. | SIFT | 0.61/0.16 | 0.55/0.16 | 0.56/0.19 | 0.53/0.19 |
| | XFeat | 0.66/0.11 | 0.63/0.11 | 0.63/0.13 | 0.59/0.14 |
| | SuperPoint | 0.73/0.13 | 0.71/0.11 | 0.71/0.15 | 0.67/0.15 |
| | | Captions similarity ($\downarrow$) | | | |
| Seg. | DPT ADE20k | 0.63 | 0.63 | 0.45 | 0.70 |
| | SAM2 | 0.61 | 0.73 | 0.49 | 0.67 |
| | GSFF | 0.63 | 0.64 | 0.55 | 0.69 |
| | SegLoc | 0.60 | 0.68 | 0.57 | 0.70 |
| SDObf. | SIFT Permutation | 0.70 | 0.78 | 0.64 | 0.71 |
| | SIFT Random Lines | 0.75 | 0.76 | 0.64 | 0.75 |
| | SuperPoint Permutation | 0.71 | 0.77 | 0.70 | 0.76 |
| | SuperPoint Random Lines | 0.72 | 0.77 | 0.69 | 0.77 |
| SDesc. | SIFT | 0.67 | 0.75 | 0.72 | 0.75 |
| | XFeat | 0.75 | 0.72 | 0.69 | 0.78 |
| | SuperPoint | 0.72 | 0.68 | 0.68 | 0.77 |

Table 3: Inversion experiments on Cambridge Landmarks. Evaluation of the privacy level of Segmentations (Seg.), Geometric Obfuscation methods (SDObf.) and Sparse Descriptors (SDesc.) through different proxies: quality of reconstructed images (with SSIM / LPIPS / FID metrics), variability of the denoising process with the same conditioning input (Mean Pairwise SSIM / Mean Pairwise LPIPS computed over 10 images reconstructed from different seeds and same conditioning) and caption similarity between LLava description of the reconstructed images and ground truth images.

| | Input | KingsCollege | OldHospital | ShopFacade | StMarysChurch |
|---|---|---|---|---|---|
| | | SSIM ($\uparrow$), LPIPS ($\downarrow$), FID ($\downarrow$) | | | |
| SIFT | FFConv (S1) | 0.52 /0.40/112 | 0.35/0.47/192 | 0.43/0.46/213 | 0.50/0.42/139 |
| | No GC (S1) | 0.53/0.30/54 | 0.38/0.27/59 | 0.44/0.33/131 | 0.49/0.33/69 |
| | Ours (S1) | 0.56/0.28/52 | 0.42/0.24/60 | 0.47/0.30/121 | 0.52/0.31/64 |
| Sparsity | FFConv (S2) | 0.49/0.47/134 | 0.33/0.50/201 | 0.41/0.53/271 | 0.48/0.51/184 |
| | FFConv (S3) | 0.44/0.56/199 | 0.30/0.56/241 | 0.35/0.61/307 | 0.44/0.62/243 |
| | Ours (S2) | 0.51/0.38/65 | 0.39/0.28/67 | 0.43/0.37/162 | 0.49/0.41/93 |
| | Ours (S3) | 0.45/0.51/90 | 0.34/0.35/82 | 0.39/0.51/214 | 0.44/0.57/134 |
| Segloc | FFConv | 0.43/0.45/144 | 0.29/0.55/236 | 0.36/0.54/323 | 0.40/0.49/214 |
| | No GC | 0.45/0.38/55 | 0.35/0.37/73 | 0.35/0.46/176 | 0.43/0.42/75 |
| | Ours | 0.47/0.37/55 | 0.35/0.36/72 | 0.38/0.44/168 | 0.45/0.40/73 |

Table 4: Inversion experiments on Cambridge Landmarks for SIFT descriptors and SegLoc segmentations. Comparison of Feed Forward convolutional approach (FFConv) vs our diffusion based approach.