# OpenReview forum: "Vulnerability of Privacy-Preserving Visual Localization against Diffusion-based Attacks"
_ICLR.cc/2026/Conference — ICLR 2026 Conference Withdrawn Submission_

### Official Review · Reviewer_cAVF · 2025-10-24

**Soundness:** 2
**Presentation:** 2
**Contribution:** 2
**Rating:** 2
**Confidence:** 2

**Summary:**

The paper introduces a method to preserve privacy for the client-side image that will be sent to the server to localize the client's position. They provide evaluation metrics for roughly two classes of existing methods: (a) sparse feature points/lines, and (b) segmentation methods. They also provide additional evaluation metrics to assess the variability across different seeds in recovering the original image from the given representation, in addition to widely used reconstruction accuracy metrics. The work incorporates diffusion-based frameworks, which are popular and among the best-performing methods in generative modeling, and LoRA as an efficient adaptation method. The results indicate that (b) segmentation-based approaches are superior in terms of prohibiting privacy breaches.

**Strengths:**

- The authors clearly describe the target applications for the proposed work.
- The proposed evaluation methods of comparing variations of different seeds are plausible.
- The paper contains a nice summary of privacy-preserving localization and mapping, and conditional diffusion models.

**Weaknesses:**

The work overlooks the complex nature of privacy protection and evaluates the difficulty of reconstructing the original image. In addition, the original scene's overall performance must be maintained, and the client should be able to handle privacy protection with minimal computational overhead. These criteria often create trade-offs between algorithmic choices, which should be further analyzed in privacy-preserving approaches. Also, the target application appears to be narrow, and the possibility of extension is not well described.

The reviewer also suggests revising the text carefully. It is hard to read some sentences, and the authors need to pay careful attention to how to put parentheses on citations, such that the sentences are complete if only the contents within the parentheses are removed.

**Questions:**

minor edits
- been introduces -> been introduced (line 117)
- Figure 1 is too small
- line 203: What are $N$ and $S$? Are they the same number?

---

### Official Review · Reviewer_3Z7k · 2025-10-31

**Soundness:** 3
**Presentation:** 3
**Contribution:** 2
**Rating:** 4
**Confidence:** 3

**Summary:**

This paper investigates the vulnerability of privacy-preserving visual localization (PPVL) systems to diffusion-based inversion attacks. The authors argue that existing approaches, such as those based on CNN inversion, are limited in their ability to reconstruct images from sparse or obfuscated representations. They propose a conditional diffusion model trained to reconstruct images from various privacy-preserving representations, including segmentation masks and sparse descriptors, using local and global conditioning mechanisms. The paper introduces a unified privacy evaluation protocol combining perceptual, structural, and semantic similarity metrics, as well as a new reconstruction diversity (RD) metric to assess uncertainty in the reconstructions. Experiments on Cambridge Landmarks and 7Scenes datasets show that diffusion-based inversion yields more faithful reconstructions than previous methods, and that segmentation-based PPVL methods are more privacy-preserving than geometric obfuscation ones.

**Strengths:**

* Addresses an important and timely topic, evaluating privacy risks in visual localization, a key component of AR/VR and autonomous systems.

* Proposes a diffusion-based privacy attack that outperforms feedforward CNN-based inversion methods, showing improved reconstruction from sparse representations.

* Introduces a unified evaluation protocol and a new metric (Reconstruction Diversity) that aim to quantify the degree of privacy preservation.

**Weaknesses:**

* Unclear threat model and attacker assumptions

The threat model is not sufficiently defined. It remains unclear what the attacker knows or can access (e.g., full access to the PPVL algorithm, model parameters, or only transmitted features). The paper assumes that the attacker already knows the type of privacy-preserving representation (e.g., segmentation or descriptors), which is a strong assumption. The authors should clarify whether the attack operates under a white-box, gray-box, or black-box scenario, as this distinction fundamentally determines the practical applicability of the attack. Without these details, the realism of the threat is difficult to assess.

* Limited scope of privacy-preserving representations evaluated

The paper tests only a few known PPVL representations (SIFT, SuperPoint, XFeat, SegLoc, GSFF, SAM2, DPT). However, in practice, many localization systems use other forms of compressed or learned representations. It is not discussed whether the proposed diffusion-based attack can generalize to other PPVL schemes, such as encrypted feature transmission or homomorphic embeddings. This omission limits the generality and impact of the findings.

* Lack of discussion on random seed influence and reconstruction variability

The proposed evaluation introduces the Reconstruction Diversity (RD) metric but does not thoroughly analyze how different seeds affect reconstruction quality or fidelity. In diffusion models, sampling variability can significantly alter perceptual and semantic content. The authors should have presented a systematic study showing how seeds impact privacy evaluation and whether reconstruction quality can be optimized or averaged across multiple runs.

* Weak connection between theoretical motivation and experimental outcomes

The motivation section emphasizes the importance of understanding what information diffusion models recover from obfuscated representations, yet the experimental results primarily focus on reconstruction quality rather than the mechanisms of leakage. More discussion linking theoretical privacy principles to observed empirical results would improve interpretability and strengthen the argument that the work contributes to the understanding of privacy risks in VL.

**Questions:**

1. What is the exact attacker’s knowledge in the proposed setup? Does the attacker have access to the PPVL algorithm, representation encoder, or only the transmitted features?

2. How sensitive are the results to random seeds during the diffusion sampling process, and can reconstruction quality be improved by averaging multiple runs?

3. Can the proposed attack generalize to encrypted or compressed representations, or does it rely on the exact structure of segmentation/keypoint data?

4. How does the diffusion-based attack compare against other generative inversion approaches, such as GAN- or autoencoder-based reconstruction, under the same conditions?

5. What are the computational costs (training and inference time) of the diffusion-based attack compared with simpler CNN inversion models?

---

### Official Review · Reviewer_4Uju · 2025-10-31

**Soundness:** 3
**Presentation:** 3
**Contribution:** 3
**Rating:** 4
**Confidence:** 4

**Summary:**

This work evaluates privacy-preserving visual localization representations through diffusion-based inversion attacks. The attack conditions diffusion models on representations via pixel-level concatenation and global graph neural network injection. Paper introduces Reconstructed Diversity measuring reconstruction variance across random seeds, and Semantic Content Recovery using vision-language models to quantify recoverable semantic information. Experiments on Cambridge Landmarks and 7-Scenes show segmentation-based methods (SegLoc, GSFF) resist reconstruction better than geometric obfuscation (Random Lines, Coordinate Permutation), evidenced by higher FID/LPIPS and reconstruction variance.

**Strengths:**

- Unified benchmark enables systematic comparison where previous PPVL methods used different evaluation protocols preventing fair assessment.
- Diffusion attack leverages probabilistic modeling to handle representation uncertainty. Table 3 shows particular effectiveness on sparse representations where feedforward networks struggle.
- Multi-scale conditioning through local concatenation and global GNN injection represents strong adversary model appropriate for privacy lower-bound evaluation.

**Weaknesses:**

- The paper's definition of privacy (p.2) seems too narrow for real-world use. It assumes "coarse semantics" (like 'residential area') don't leak privacy, but in many applications (like geography or surveillance), this "coarse" arrangement of landmarks is precisely what enables identification.
- The analysis of segmentation granularity (Table 1) is confusing. It's unclear if the better reconstruction from SAM2 is due to finer granularity or image-specific masks (vs. DPT's general categories). These two factors are confounded, making the conclusion about granularity difficult to trust.
- The Semantic Content Recovery (SCR) metric relies on a single VLM. Since different VLMs (like CLIP vs. LLaVA) capture semantics differently, it's hard to know if this result is robust or just an artifact of the chosen model.

**Questions:**

- How should the privacy definition adapt to different contexts? For geographic or surveillance applications, shouldn't coarse semantics also be considered sensitive?
- Could the authors design an experiment to isolate the effect of granularity? For example, by progressively merging SAM2 masks to see how reconstruction quality changes.
- Does the Semantic Content Recovery ranking (segmentation > geometric) hold true if tested with other VLMs that capture semantics differently?

---

### Official Review · Reviewer_1zDS · 2025-11-02

**Soundness:** 3
**Presentation:** 3
**Contribution:** 2
**Rating:** 4
**Confidence:** 4

**Summary:**

The problem of privacy-preserving visual localization is considered. The paper focused on the vulnerability of current privacy-preserving representations to inversion attacks. The authors introduce a novel diffusion-based privacy attack that trains a diffusion model to reconstruct images by conditioning on various privacy-preserving representations, serving as a proxy to quantify the amount of sensitive information leaked. The paper provides a comparison across multiple geometric and descriptor obfuscation methods for privacy-preserving visual localization. The findings show that segmentation-based representations offer a better degree of privacy than geometric obfuscation methods.

**Strengths:**

1. This work proposed a new diffusion-based attack/inversion, where a diffusion model is trained to recover the original image from a privacy-preserving representation (e.g. local descriptors, segmentation maps).
2. A new evaluation protocol based on the robustness of privacy representation to diffusion-based inversion.

**Weaknesses:**

1. Lack of comparison to state-of-the-art baselines (for descriptor-based representations). It is already shown by Chelani et al. (2025) that Obfuscation based representation is vunlerable to inversion attack. Why is the introduction of diffusion-based attack important?
2. For segmentation-based representations, the reconstruction quality seems quite bad both quantitatively and qualitatively. As far as I can tell, it is safe to conclude that the segmentation-based representation is robust the diffusion-based inversion, which adds to my concern about the importance of the proposed diffusion-based attack.
3. A privacy evaluation protocol based on robustness to certain attack is useful only if the considered attack is a serious and special threat, which remains to be validated in the case of the proposed attack.

**Questions:**

1. What are the visual localization performances for the privacy-preserving representations in Table 2? For fair comparisons between different representations, the privacy-utility tradeoff has to be taken into consideration. A non-informative representation can be perfectly privacy-preserving but useless for downstream application.

---

### Note · Authors · 2025-11-12

I have read and agree with the venue's withdrawal policy on behalf of myself and my co-authors.